# Surgical Management of Pediatric Coronoid Process Fractures: A Report of Two Cases

**DOI:** 10.3390/life15040614

**Published:** 2025-04-06

**Authors:** Anna Gabriella Lamberti, Aba Lőrincz, Tibor Molnár, Tamás Kassai, Hermann Nudelman, Gergő Józsa

**Affiliations:** 1Division of Surgery, Traumatology, Urology, and Otorhinolaryngology, Department of Pediatrics, Clinical Complex, University of Pécs, 7 József Atilla Street, 7623 Pécs, Hungary; lambi.anna.gabii@gmail.com (A.G.L.); aba.lorincz@gmail.com (A.L.); molnar.tibor97@gmail.com (T.M.); jozsa.gergo@pte.hu (G.J.); 2Department of Pediatric Traumatology, Péterfy Hospital, Manninger Jenő National Trauma Center, 17 Fiumei Street, 1081 Budapest, Hungary; kassai.tamas@obsi.hu

**Keywords:** pediatric, coronoid, ulna, fracture, pelvic autograft, coronoid fracture, elbow, trauma

## Abstract

Coronoid process fractures in the pediatric population are rare and often misdiagnosed, leading to chronic elbow instability. We aim to evaluate the surgical management of two adolescent cases of inveterate coronoid fractures using autologous bone grafting. Both patients, with a history of recurrent elbow dislocations, presented with pseudoarthrosis and were initially misdiagnosed due to minor or subtle fractures. Comprehensive imaging, including computed tomography (CT) and magnetic resonance imaging (MRI), confirmed the presence of significant coronoid defects. The surgical intervention involved employing autografts from the iliac wing to reconstruct the coronoid process, followed by fixation with screws. Both patients underwent postoperative rehabilitation via physiotherapy, resulting in full functional recovery. At their one-year follow-ups, both patients regained full elbow function, achieving range-of-motion measurements of 0–0–130° flexion–extension and 90–0–90° pronation–supination; no recurrence of instability was reported, with no complications at the yearly follow-ups. This approach demonstrates the efficacy of autograft reconstruction in restoring elbow stability, particularly in cases with substantial bone loss or pseudoarthrosis. Our study highlights the importance of advanced imaging and individualized treatment strategies, emphasizing that early surgical intervention can prevent long-term disability in pediatric patients with chronic coronoid fractures.

## 1. Introduction

Coronoid fractures are oftentimes encountered alongside other injuries of the elbow, particularly dislocations and ligamentous injuries [1]. These fractures are more common in adults, and their incidence among children is fairly low, ranging from 3 to 6% of all pediatric elbow fractures in the literature [1,2,3]. Within this subset, avulsion injuries ensue at an estimated rate of around 3.5% and are associated with high-energy trauma. The coronoid process is a pivotal bony prominence forming part of the sigmoid notch—together with the olecranon—where it articulates against the distal humerus [1,4]. Functionally, this structure serves as an anterior buttress, preventing the posterior displacement of the ulna, and also provides an important insertion point for the anterior bundle of the ulnar collateral ligament, thus contributing to valgus stability [1,2,3].

In the pediatric population, where small fragments may be subtle on plain radiographs, coronoid fractures are often overlooked or misdiagnosed despite their crucial role in maintaining elbow stability. The vast majority of elbow dislocations are posterior, about 95%, emphasizing the coronoid process’s critical function in preventing posterior subluxation or dislocation. When these fractures occur combined with ligamentous laxity or associated osseous injuries, elbow instability can become pronounced, necessitating a comprehensive assessment [1,5,6].

X-rays remain the first-line imaging modality for suspected coronoid fractures; however, their sensitivity is reduced in detecting small fragments in the pediatric population, especially in cases with subtle fractures. Therefore, computed tomography (CT) imaging is often used for diagnosis and surgical planning [7]. The problem with non-ionizing imaging options, such as ultrasound (US), is that their interpretation demands considerable expertise. Magnetic resonance imaging (MRI), although effective in providing detailed soft tissue evaluations, is time-consuming and may necessitate general anesthesia in younger children due to the small space and still positioning required [7,8].

Several classification systems have been developed, such as the O’Driscoll, Regan and Morrey, and Mayo classifications; however, they fail to account for complex or combined injury patterns [6,9]. Fractures associated with concurrent soft tissue or ligamentous injuries can alter treatment decisions significantly, yet these classifications do not address these intricacies [2,10]. Therefore, treatment approaches must be based on the severity of the fracture, associated injuries, and the child’s age. Conservative management may be adequate for non-displaced fractures, with continuous close monitoring for signs of instability. Surgical intervention, most commonly open reduction and internal fixation (ORIF), is indicated for displaced or unstable fractures [4,9]. In more complex cases, surgical fixation with ligament reconstruction may be necessary [1,4].

This study strives to illustrate the management of complex scenarios by presenting two cases. By exploring the diagnostic process, surgical technique used, and rehabilitation given, we intend to highlight challenges and provide insight into potential solutions that may guide future clinical decision-making.

## 2. Case Reports

### 2.1. Case 1

An 11-year-old girl reported to our infirmary on the 10 October 2020 with recurrent posterior subluxation of the elbow. The patient’s history described a previous injury in 2017 when she jumped from a swing and ended up injuring her left elbow (Figure 1). This luxation injury was reduced and treated conservatively with a cast for six weeks. In 2018, she suffered another injury on the same extremity, which was treated the same way. In April 2019, her elbow dislocated three times in a single day, so a cast was applied for seven weeks, and X-rays and CT scans were performed.

In 2020, her elbow dislocated once again when she was climbing onto her bed. In 2020, on the 1 October, she suffered a fall after she was pushed during a sports activity but had only pain and no dislocation, so she received a splint. A few days later, on the 7 October, spontaneous subluxation of the elbow occurred while the child was getting dressed. We applied a dorsal cast to prevent the extremity from suffering further dislocations and planned further static and dynamic diagnostic tests to evaluate the cause of the recurrent dislocations, as the root cause could not be identified on the initial CT or X-ray images that were provided in 2019 (Figure 2).

Diagnostic stability tests were carried out before and after reconstructive surgery, and, until then, an orthosis was provided to prevent the subluxation of the arm. Diagnostic tests were carried out prior to the planned operation under the influence of inhalation anesthesia. These included valgus stress in the 0- (extended), 30-, 60-, and 90-degree (°) positions, to which the medial ligamentous structures responded well by not letting the joint open or be subluxated. In the extended and 90° positions, the dorsally oriented stress test proved stable; however, in the 60° position during flexion, instability was noted, in which the joint was undoubtedly subluxated (Figure 3) while the summary of events can be found in Table 1.

Based on these results and with the help of CT imaging for surgical planning, a ventral stabilizing operation was planned, with autograft from the iliac crest. The indications for surgery included several important points, such as the recurrence of subluxation, the formation of pseudoarthrosis, and the diagnostic confirmation of anatomical irregularity.

The surgery took place after disinfection and isolation, under general anesthesia, in exsanguinated conditions, and with prophylactic antibiotic protection (1 g Cefazolin) and took 90 min to complete. We used a ventral longitudinal incision along the anterior aspect of the elbow, following the modified Henry approach. After incising the skin and subcutaneous tissue, the bicipital aponeurosis was identified and carefully incised to expose the underlying brachial artery and median nerve, which were mobilized and retracted medially for protection. The biceps tendon was retracted laterally to enhance visualization. The brachialis muscle was then split longitudinally in line with its fibres, allowing access to the anterior joint capsule. Once the capsule was incised, the pronator teres was identified along its ulnar attachment and gently retracted, providing direct exposure to the coronoid process for reconstruction, facilitating precise debridement and accurate graft placement. The 2 × 1.5 × 3 millimetre (mm) bone fragment was removed. Histological analysis did not identify underlying pathology and proved that the fragment was otherwise formed of healthy bone tissue. The autograft was obtained from the iliac ala. From here, a 1 × 1 × 1 centimetre (cm) corticospongious piece was borrowed and reattached to the refreshed spongious surface of the coronoid process with two, one 26 and one 28 mm in length, A-spire (Sanatmetal Ltd., Eger, Hungary) screws (Figure 4). The graft size was determined based on the intraoperatively measured defect and used to restore anatomical height and contour to ensure joint congruity and stability. The recipient site was refreshed to gain access to a fresh vascular supply for graft incorporation with the help of a Volkmann spoon. A drain was applied, and after the reconstruction of the superficial layers, the skin was sutured. No additional reconstruction or reinforcement was needed for the median collateral ligament (MCL), as both the anterior and posterior maintained proper ligamentous integrity during the stability test. After swathing and bandaging, a 90° cast was applied, which was utilized for one month.

During this time, the elbow was immobilized and the fingers were actively moved multiple times a day. The initial healing period progressed without any complications, numbness, pain, or excess swelling. After cast removal, aqua therapy was initiated, with physiotherapeutic exercises performed twice daily for the following 14 weeks. Two months later, the scar was healed with a function identical to the opposing side. On control X-ray imaging, the graft could be clearly visualized with adequate positioning. At the half-year mark, the graft was incorporated, which could be seen from the X-ray images, with a function of 0–0–130° in flexion–extension and 90–0–90° in pronation–supination. After one year, the range of motion (ROM) did not decrease, and the child was able to carry out gymnastic exercises with ease, as well as sports activity in the form of handball’ the child remains symptom- and complication-free. A yearly follow-up examination will be performed until adolescence and should be conducted uniformly across all visits.

### 2.2. Case 2

The following case of elbow fracture has a long history that includes several surgical managements. The patient was a 14-year-old boy who had suffered multiple elbow injuries in the past. At the age of four, he suffered a left supracondylar humerus (SCH) fracture with a large displacement (Gartland IV). Closed reduction and osteosynthesis with Kirschner (K-) wires were performed. The wires were removed five weeks later.

At the 3rd month post-op follow-up, an extension deficit and ulnar nerve dysfunction of the little and ring fingers were noted. One month of intensive physiotherapy restored normal function. At age six, he sustained another left-arm SCH fracture without displacement, which healed clinically in three weeks with conservative treatment.

When he was 12, due to a left medial epicondyle fracture with a dislocation of two mm, a long splint was placed on the upper extremity for three weeks. After the removal of the splint, the movements of the elbow were narrowed.

At the age of 13, another injury occurred, and the child complained of a sprain prior to the injury. X-ray imaging showed a posterior elbow dislocation without signs of a fracture (Figure 5). Under anesthesia, the luxation was repositioned. Following repositioning, an image review was performed, and no suspected fracture area was identified. A long upper limb cast was placed for two weeks.

He suffered another elbow trauma due to a fall at the age of 14, resulting in pain and dislocation. X-rays showed no recent fracture, but a 5 mm rounded bone fragment was visible. MRI was requested for diagnostic purposes and further exploration (Figure 6). The results indicated that the fragment originated from the coronoid process and that the elbow was in a dorsally subluxated position. With regard to the instability, the exploration of the joint with debridement and reconstruction was proposed. A CT scan was performed for surgical planning, which yielded equivalent results to the MRI. A summary of events can be seen in Table 2. 

The surgery took place after disinfection and isolation, under general anesthesia, in exsanguinated conditions, and with prophylactic antibiotic protection (1 g Cefazolin) over 90 min. During surgery, the left elbow joint was accessed from an anterior incision, utilizing the modified Henry approach once again. The detached bone fragment was removed and replaced with a corticospongious block obtained from the iliac ala (Figure 7). Preparation of the recipient bed at the coronoid defect site was performed by removing fibrous tissue and refreshing the surface to expose healthy, bleeding cancellous bone. Subsequently, the corticospongiosal autograft was harvested from the ipsilateral iliac ala. The graft was carefully shaped intraoperatively to match the defect dimensions accurately, ensuring an optimal fit to restore the anatomical contour of the coronoid process. This technique aimed to facilitate biological integration and promote rapid healing. The graft was fixed in its final position with two Herbert screws (Johnson & Johnson: Depuy Synthes, Oberdorf, Switzerland). A drain was applied, and after the repair of the subcutaneous layers, the skin was sutured. Following swathing and bandaging, the left upper extremity was immobilized with a long upper limb cast.

Postoperative X-rays showed the adequate position of the corticospongiosal block. After three weeks, the cast was removed and physiotherapy was initiated. After five weeks, the child complained about numbness in his ring and little fingers. At that time, his ROM was slightly limited during flexion and extension; however, his pronation–supination range was complete. Physiotherapy was continued to better the ROM of the elbow. By the third month, the joint movements were complete, with no signs of elbow instability. Control X-rays revealed signs of proper graft remodelling. By the eighth month, the fracture was healed clinically and radiologically (Figure 8).

From the X-ray imaging, it was seen that the fracture was rebuilt and the coronoid process was stabilized in a good position, with a complete ROM in the elbow joint (Figure 9).

## 3. Discussion

The diagnostic challenge stems from the rarity of the occurrence of these cases, especially when the fragment is small or when primary imaging fails to detect early signs of instability. Both cases illustrate how significant coronoid pathology can manifest with minimal findings on imaging, underscoring the importance of recurrent subluxations. These repeated events highlight the significance of thorough imaging (including CT and/or MRI) and locating subtle fractures or pseudoarthroses through dynamic testing.

Though part of the standard imaging technique, plain film radiographs can be insufficient in detecting small fragments or anatomical irregularities of the coronoid [2,11]. In our first patient, stability tests under anesthesia were carried out, revealing instability at around 60° of flexion. In the second patient, after multiple injuries, which were managed conservatively, an MRI and subsequent CT confirmed a 5 mm fragment. These cases stress the importance of advanced imaging, which is essential to confirm the diagnosis and guide treatment, especially when clinical suspicion remains high despite negative or inconclusive radiographs.

The management approach depends on the fragment size, degree of displacement, presence or absence of concomitant ligamentous or cartilaginous injury, and elbow instability [1,7,12]. Growth plate considerations further complicate decision-making in the pediatric population. Conservative management is indicated in the case of small, minimally displaced fragments with stable elbow mechanics and no significant ligamentous involvement. These require short-term immobilization (2–4 weeks) in a cast or splint and transition to early ROM exercises. This is often successful in children due to their robust healing potential and ability to remodel minor defects [8].

ORIF is indicated for larger or significantly displaced fragments (e.g., Regan and Morrey Type II or III) and associated fracture dislocations or joint instability. Most commonly, an anterior approach to the elbow is used for direct visualization of the coronoid and its subsequent fixation with small screws (e.g., Herbert screws, cannulated screws) [6,7,8]. Suture anchors can be employed in certain avulsion or osteochondral fractures. This restores the coronoid’s function as an anterior buttress. Furthermore, it minimizes the risk of recurrent posterior subluxation or dislocation.

Suture or anchor fixation can be utilized in cases of tiny or osteochondral fragments not amenable to screw fixation or avulsion-type fractures [13]. This technique includes the placement of suture anchors or heavy sutures that approximate and secure the fragment. The method is effective in preserving articular congruity and may be particularly useful in younger children with smaller bony fragments [13].

Chronic elbow instability, rather than the isolated fracture itself, often dictates the management strategy used for coronoid injuries. Treatment should be guided by the severity of the instability, any associated ligamentous injury, and the chronicity of symptoms rather than the fragment size alone. Conservative treatment can manage minor, stable injuries effectively. However, recurrent instability, particularly chronic instability, generally necessitates surgical reconstruction. In such scenarios, open reduction and internal fixation (ORIF) combined with soft tissue repair or ligament reconstruction is advocated for, aligning with recommendations from the literature on adult fractures. Reconstruction often involves autograft procedures using corticospongiosal bone grafts, which have demonstrated superior outcomes in restoring anatomical alignment, stability, and function in chronic elbow instability cases [1,11,13,14]. Additionally, incorporating ligament repair or reconstruction simultaneously is advised when persistent instability is confirmed clinically.

In instances where a significant bony defect, a nonviable fragment, or an established pseudoarthrosis is present—such as seen in both of our cases—an autologous bone graft (autograft) can be critical to restoring the native height and outline of the coronoid [11]. This differs from simply fixing the existing fragment; in chronic cases, autologous grafting is often preferred over other surgical techniques due to its superior biological integration, rapid healing potential, and its ability to restore anatomical congruity and joint stability effectively. Unlike synthetic substitutes or allografts, autologous grafts have osteogenic, osteoinductive, and osteoconductive properties, facilitating better graft incorporation and reducing complications related to graft rejection or failure. Recent studies support these advantages, demonstrating improved outcomes in terms of functional restoration and long-term stability with autograft reconstruction [11,13,14].

In comparison, traditional ORIF without grafting is sufficient for simple fractures with reducible fragments. Non-operative techniques are the most successful for stable, minimally displaced fractures, while anchor fixation works well for undersized avulsions. However, once significant bone loss or chronic pseudoarthrosis is present—as observed in both of our cases—autograft reconstruction becomes a more ideal solution for restoring elbow stability.

The long-term prognosis of pediatric coronoid fractures, especially those requiring surgical intervention and autograft reconstruction, is generally favourable. However, clinicians must remain vigilant about potential complications. Growth disturbances, although rare, can manifest as deformities or alignment discrepancies due to injury or surgical manipulation near growth plates. Late elbow instability or arthrosis may also arise if anatomical congruity and joint biomechanics are inadequately restored or if rehabilitation protocols are insufficiently implemented. Regular clinical and radiographic monitoring until skeletal maturity is crucial to promptly detect and manage these issues, thereby ensuring optimal functional outcomes and minimizing complications. This approach aligns with recent pediatric and adult research that emphasizes the importance of careful long-term surveillance post-reconstruction [13,14].

Generally, if recognized and treated appropriately, coronoid fractures—even those requiring grafting—tend to heal well, with children often regaining full elbow motion. Radiographic monitoring until skeletal maturity is advised to look for complications such as growth disturbances, graft incorporation, and potential late elbow instability or arthrosis. Reconstruction may reliably address large bony defects or chronic nonunions, ensuring stable outcomes for patients whose recurrent subluxations might otherwise lead to permanent functional deficits.

This study is limited by its small sample size and its retrospective nature, which restricts its generalizability. Additionally, there was no control group or standardized protocol for preoperative imaging and postoperative rehabilitation, which may affect the reproducibility of our results. Further, large-scale prospective studies with comparative designs are necessary to validate these observations and refine treatment strategies for infrequent injuries.

## 4. Conclusions

The missed diagnosis of these fractures in adolescents may lead to chronic elbow instability if left untreated. Coronoid reconstruction with a bone graft is a viable option even in pediatric patients, providing excellent functional outcomes. When combined with rigid internal fixation and a structured rehabilitation programme, this approach can effectively prevent recurrent subluxations and long-term functional impairment, even in chronic or recurrent cases.

## Figures and Tables

**Figure 1 life-15-00614-f001:**
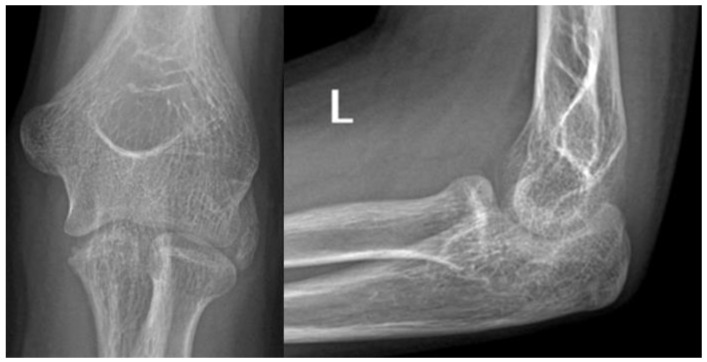
Plain radiographs taken at the initial injury.

**Figure 2 life-15-00614-f002:**
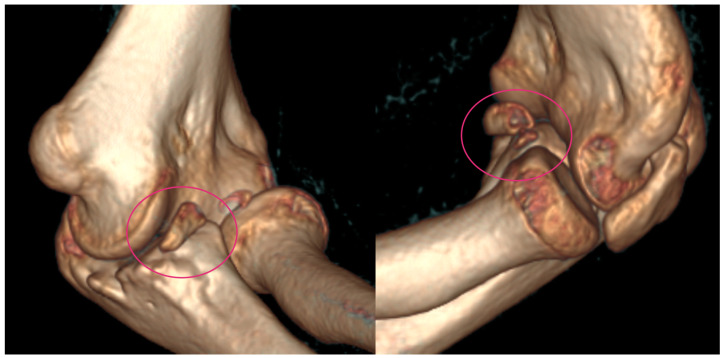
CT images were provided for surgical planning; the defect can be seen highlighted in the pink circle.

**Figure 3 life-15-00614-f003:**
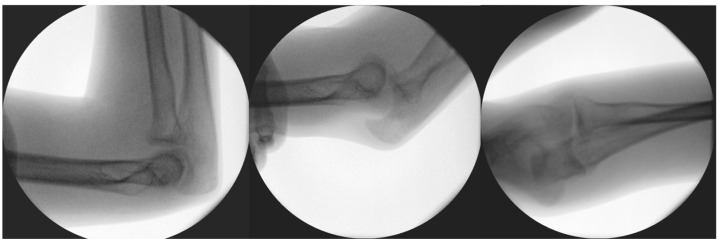
Results of diagnostic dynamic testing under fluoroscopy.

**Figure 4 life-15-00614-f004:**
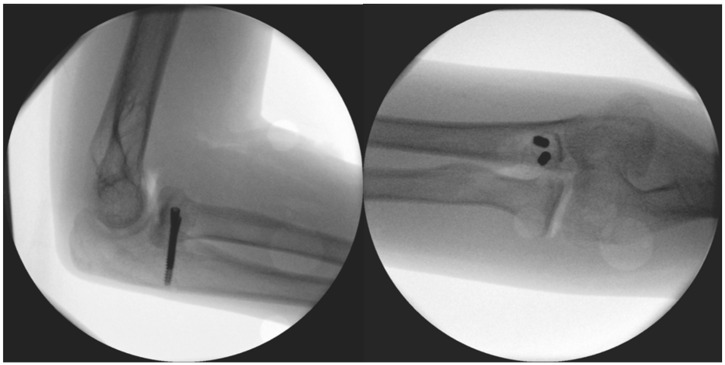
Immediate post-op images of the fixation.

**Figure 5 life-15-00614-f005:**
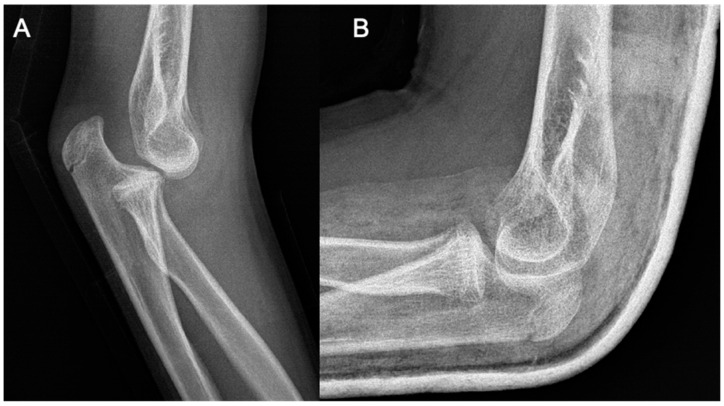
Imaging of the posterior dislocation at age 13, image (**A**) showing the luxation upon ebow extension and (**B**) from a lateral view in flexed position.

**Figure 6 life-15-00614-f006:**
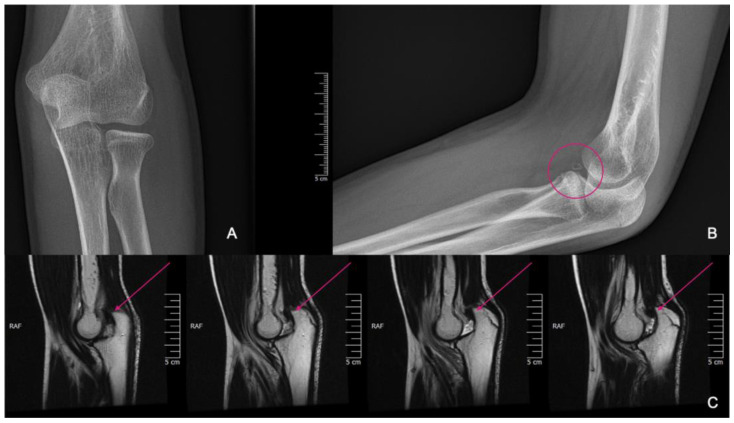
X-rays showing fragmentation, highlighted by the pink circle (**A**,**B**). Subluxation is visible on the MRI pointed at with pink arrows (**C**).

**Figure 7 life-15-00614-f007:**
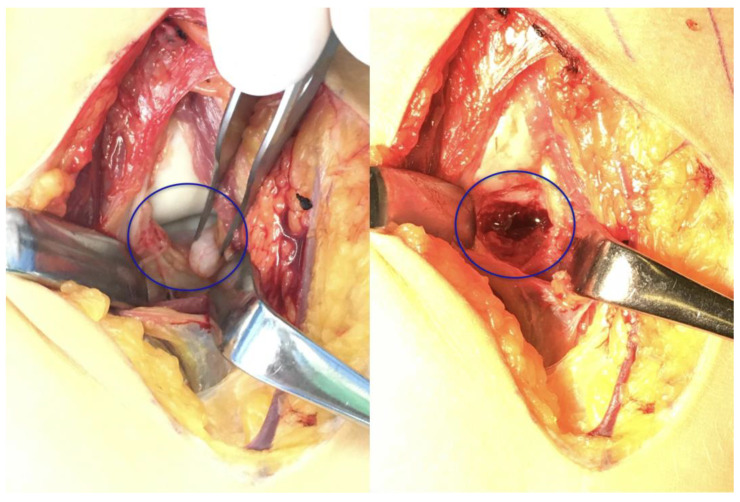
Intraoperative images highlighting the separate bone fragment and the defect (blue circle).

**Figure 8 life-15-00614-f008:**
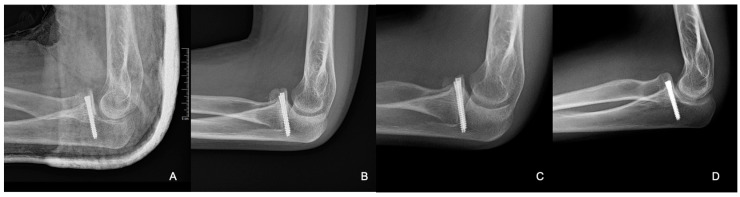
Radiographic control images from weeks 3 (**A**), 8 (**B**), 24 (**C**), and 40 (**D**).

**Figure 9 life-15-00614-f009:**
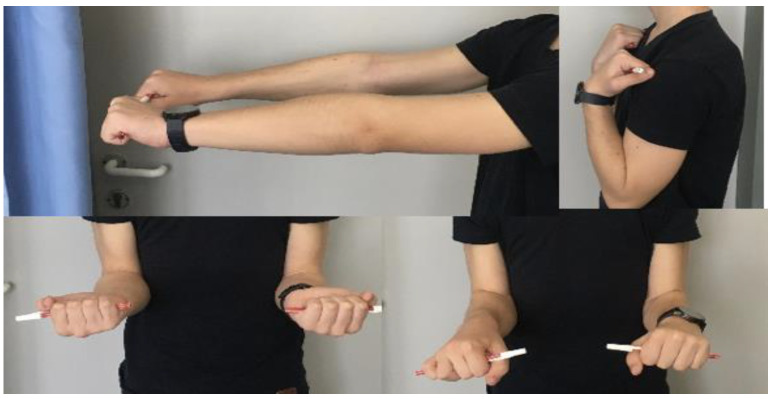
Functional images ten months after the reconstruction.

**Table 1 life-15-00614-t001:** Timeline of events for case 1.

Date/Time	Event/Clinical Intervention	Outcome/Status
2017	Initial elbow injury, posterior luxation	Conservative management, cast (6 weeks)
2018	Recurrent injury	Conservative management, cast
2019 April	Multiple dislocations	Conservative management, cast (7 weeks), CT/X-ray imaging
2020 October	Spontaneous recurrent subluxations	Instability confirmed, surgery planned
2020 October (Surgery)	Coronoid reconstruction (iliac autograft + screws)	Successful surgical fixation
2020 November	Immobilization (4 weeks), rehabilitation initiated	Physiotherapy and aqua therapy started
2021 April (6 months)	Follow-up evaluation	Full ROM, graft incorporated
2021 October (1 year)	Follow-up evaluation	Symptom-free, full ROM, returned to sports
Annually thereafter	Planned follow-up until skeletal maturity	Monitor stability, growth, and joint function

**Table 2 life-15-00614-t002:** Timeline of events for case 2.

Age/Timepoint	Event/Clinical Intervention	Outcome/Status
Age 4	Supracondylar humerus fracture	Surgical K-wire fixation, healed
Age 6	Recurrent supracondylar humerus fracture	Conservative treatment (cast immobilization)
Age 12	Medial epicondyle fracture	Splint (3 weeks), functional limitations
Age 13	Posterior elbow dislocation	Closed reduction, conservative treatment
Age 14	Elbow dislocation with chronic instability, coronoid fracture identified	Surgical reconstruction (iliac autograft, Herbert screws)
3 weeks post-op	Cast removal and rehabilitation initiated	Physiotherapy started
3 months post-op	Follow-up evaluation	Complete ROM, stable elbow
8 months post-op	Follow-up evaluation	Graft integration
Annually thereafter	Planned follow-up until skeletal maturity	Monitoring stability and functional outcomes

## Data Availability

The data are contained within this article.

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
