# Peer review of "Surgical Management of Pediatric Coronoid Process Fractures: A Report of Two Cases"

_life, 2025, doi:10.3390/life15040614_

Round 1
Reviewer 1 Report
Comments and Suggestions for Authors
Comments are in the attached article

needs to be improved.
Reviewer 2 Report
Comments and Suggestions for Authors
Thank you for this interesting manuscript about two pediatric cases of inveterate coronoid process fractures managed surgically using autologous iliac crest bone grafting, with an emphasis on advanced imaging and dynamic testing to guide treatment. The report aims to demonstrate that early surgical intervention with graft reconstruction can restore elbow stability and prevent long‐term disability in this rare and challenging injury pattern.
The topic is clinically relevant given the rarity and diagnostic challenges of pediatric coronoid fractures, and the manuscript offers valuable insights into a surgical strategy for chronic cases. However, there are several issues regarding clarity, methodological detail, and literature integration that require substantial revision before the manuscript can be considered for publication in your journal.
Abstract: provides a clear overview of the clinical problem, surgical intervention, and outcome. Mentions the use of advanced imaging and postoperative rehabilitation. Quantitative details (e.g., specific range-of-motion values, duration of follow-up) could be added to strengthen the impact
Introduction: provides adequate background on the incidence and significance of coronoid fractures in the pediatric population and emphasizes the diagnostic difficulties with standard imaging modalities. Please, expand on why autologous grafting is preferred over other surgical techniques in these inveterate cases. Some citations are outdated; incorporating more recent literature could enhance relevance.
Case Reports detailed patient history and imaging findings are presented. The surgical technique is described with specifics regarding graft dimensions and fixation methods. Figures should be of high resolution with clear labeling; if available, include a timeline summarizing the clinical course. Discuss potential complications and how they were mitigated or monitored, even if not encountered.
Discussion: reiterates the importance of advanced imaging and dynamic testing in the diagnosis of subtle pediatric coronoid fractures. Conservative versus surgical management are compared and highlights the advantages of autograft reconstruction. Please address the long-term prognosis, including potential growth disturbances or late instability, in greater depth.
Conclusions: summarizes the key finding that autologous bone grafting can restore elbow stability in chronic pediatric coronoid fractures. Clear future directions are suggested (e.g., prospective studies or multicenter case series) to validate the findings.
General statement:
I recommend that the manuscript be reconsidered after minor revision. The topic is important and the cases are interesting; however, significant improvements in clarity, detail, and critical discussion are necessary before the manuscript meets the standards of your journal.
English is fine
Round 2
Reviewer 1 Report
Comments and Suggestions for Authors
Please check the comments on the article

Needs some refinement
